# Rescue of ApoE4-related lysosomal autophagic failure in Alzheimer's disease by targeted small molecules

Meenakshisundaram Balasubramaniam [1✉], Jagadeesh Narasimhappagari[1], Ling Liu[1], Akshatha Ganne[1], Srinivas Ayyadevara[1,2], Ramani Atluri[1], Haarika Ayyadevara[3], Guy Caldwell [4], Robert J. Shmookler Reis[1,2], Steven W. Barger[1,2] & W. Sue T. Griffin [1,2✉]

Homozygosity for the ε4 allele of *APOE* increases the odds of developing Alzheimer's by 12 to 15 times relative to the most common ε3;ε3 genotype, and its association with higher plaque loads comports with evidence that *APOEε4* compromises autophagy. The ApoE4 protein specifically binds a *cis* element ("CLEAR") in the promoters of several autophagy genes to block their transcription. We used a multifaceted approach to identify a druggable site in ApoE4, and virtual screening of lead-like compounds identified small molecules that specifically bind to this site to impede ApoE4::DNA binding. We validated these molecules both in vitro and in vivo with models expressing ApoE4, including ApoE4 targeted-replacement mice. One compound was able to significantly restore transcription of several autophagy genes and protected against amyloid-like aggregation in a *C. elegans* AD model. Together, these findings provide proof-of-principle evidence for pharmacological remediation of lysosomal autophagy by ApoE4 via ApoE4-targeted lead molecules that represent a novel tack on neurodegenerative disorders.

[1] Department of Geriatrics, University of Arkansas for Medical Sciences, Little Rock, AR, USA. [2] Central Arkansas Veterans Healthcare System, Little Rock, AR, USA. [3] University of Arkansas, Fayetteville, Fayetteville, AR 72701, USA. [4] University of Alabama, Tuscaloosa, AL 35487, USA.
✉email: mbalasubramaniam@uams.edu; GriffinSueT@uams.edu

Among the multiple risk factors associated with the development of Alzheimer's disease (AD), foremost is inheritance of one or both ε4 alleles of the apolipoprotein gene (*APOE*). Among Caucasians, inheritance of two *APOE*ε4 alleles (genotype *APOE*ε4,4) confers a risk for development of AD 12 to 15 times that of *APOE*ε3,3; even inheritance of one allele increases risk by 4-times[1]. Further, several studies report that compared to *APOE*ε3,ε3 carriers, AD patient carriers of *APOE*ε4,ε4 have conspicuous elevations in AD-associated hall-mark pathognomonic aggregates of AD, *viz.*, Aβ plaques and neurofibrillary tangles[2]. These reports are consistent with an *APOE*ε4-related failure in lysosomal autophagy, the cellular mechanism for disposal of such large aggregates as well as other large entities such as defective mitochondria. As evidence, Simonovitch and colleagues[3] found that in the presence of ApoE4 autophagy is impaired and is associated with reduced clearance of Aβ aggregates. Similarly, we provided evidence of an ApoE4-related increase in aggregates in a human cell culture model, which mirrored a marked diminution in transcription of three lysosomal autophagy-dependent transcription factor-EB (TFEB)-regulated mRNA transcripts (*SQSTM1, LAMP2,* and *MAP1LC3B*) in brain tissues from AD patient carriers of *APOE*ε4,4 compared to those from AD *APOE*ε3,ε3 carriers[4]. Using a multifaceted approach of combining computational modeling and molecular assays, we showed that the ApoE4 protein competes with TFEB to directly and specifically bind to CLEAR (Coordinated Lysosomal Expression and Regulation) DNA motifs and thus impedes TFEB-mediated transcription of the genes required for lysosomal autophagy. Further support for our findings comes from studies by Lima et al. [5] using differential pulse voltammetry (DPV) and electrochemical impedance spectroscopy (EIS) to show that ApoE4 binds a CLEAR DNA sequence much more avidly than do ApoE2 or -3. Further, using protein-DNA docking and molecular dynamic simulation studies, we predicted three-dimensional models implying direct interactions between ApoE4 and CLEAR DNA motifs at the atomic level[4].

Based on our previous report[4] and empirical data-driven models presented here, we hypothesized that targeting with small molecules that bind exclusively to the DNA-binding region of the ApoE4 protein would block ApoE4 interactions with CLEAR DNA motifs and in this way restore transcription of the essential genes, including *SQSTM1, MAP1LC3B,* and *LAMP2,* which are required for efficient clearance of pathognomonic aggregates via lysosomal autophagy. In the work reported here, combining computational and experimental approaches, we identified several small molecules that preferentially target ApoE4 over ApoE3. Each of these small molecules was tested in multiple ApoE4-expression models for efficacy in restoring TFEB-mediated autophagic transcription, and one was shown to restore expression of three critical autophagy-related genes, affording promise of restoration of lysosomal autophagy to that found in individuals with genotype *APOE*ε3,3.

## Results

### Molecular-dynamic simulation predicts a druggable pocket in ApoE4
Our recent findings in brain tissue from AD patients—which show that ApoE4, but not ApoE3, competes with TFEB to suppress lysosomal autophagy by directly binding to the CLEAR-DNA promoter site[4]—led us to posit that a small molecule that stably binds ApoE4 would disrupt its interaction with CLEAR-DNA motifs. Such molecular inhibitors would allow normal TFEB CLEAR-DNA binding to drive the transcription of mRNAs that encode proteins required for lysosomal autophagy.

Our strategy was to first identify key ApoE4 amino-acid residues that interact with the CLEAR-DNA motif. The three-dimensional structure of ApoE4 protein, which we previously modeled and reported as comprising chiefly weak α-helices and random coils (Fig. 1a)[4], qualifies as a partially disordered protein. Because the structural dynamics tend to be unstable for any disordered protein, their targeting with small molecules is challenging (Fig. 1). This may be especially true in targeting the DNA-binding aspect of ApoE4 with the goal of disrupting ApoE4 interaction with CLEAR motifs in DNA (Fig. 1b). Protein-DNA interaction analysis predicted a DNA-binding interface of ApoE4 that contains several arginine residues and has preferential affinity for CLEAR motifs. In particular, Arg112 is unique to ApoE4 and distinguishes it from ApoE3[4,6]. Further, this Arg112-containing sequence directly interacts with CLEAR-DNA motifs (Fig. 1c). Based on these predictions, in order to impede ApoE4-DNA interactions, we sought to find a druggable pocket that has a stable conformation and is near to or at the ApoE4 DNA-binding region (Fig. 1b-c). For this, we performed 300-ns atomistic molecular-dynamic (MD) simulations using the Desmond simulation package[7]. Simulation trajectory analysis indicated that, despite the disordered regions at the N- and C-terminal regions of ApoE4, the overall structure, including the DNA-binding region, was dynamically stable. Root Mean Square Deviation (RMSD) of 300-ns simulation trajectories showed a stable plateau, indicating a stable conformation in ApoE4 (Fig. 1d). To further assess the stability of the ApoE4 structure, we analyzed the number of internal H-bonds in ApoE4 from the 300-ns simulation trajectories. Results show that the number of internal H-bonds did not vascillate and remained stable throughout the 300-ns simulation (Fig. 1e), indicating that the overall structure of ApoE4 did not unfold but instead remained stable. Collectively, these findings indicate that the dynamic structure of the ApoE4 DNA-binding region is sufficiently stable and therefore can be analyzed for the presence of a druggable pocket as a target for small molecules, which might serve as drugs to prevent ApoE4::CLEAR DNA-binding.

### Small molecule binding-site prediction identifies a druggable target in ApoE4
Atomistic molecular dynamics simulation revealed a stable conformation between two helical domains of the DNA-binding region (Fig. 1c). Based on the average RMSD, we selected a structural conformation of ApoE4 within the RMSD stable plateau as our initial structure for drug targeting. We used Binding Site Prediction tools from the BIOVIA Discovery Studio package to predict a potential ligand-binding pocket(s) in the ApoE4 structure. We identified 7 receptor cavities (druggable pockets) in ApoE4, of which the first two sites are located at the far end of the DNA-binding region, where they arise between the highly fluctuating coils of this protein. A third site, however, is found near the center of the ApoE4 DNA-binding region (Fig. 2a, green mesh), in which the helical structures inferred from MD simulation analysis are more stable than other predicted sites within the ApoE4 protein, thus favoring this site for targeting small molecules to impede the DNA-binding potential of ApoE4 (Fig. 2a, b). Amino-acid-residue analysis shows that this predicted druggable pocket in the ApoE4 protein encompasses residues that are crucial for ApoE4 binding to CLEAR motifs, including Arg61, Arg172, Arg178, and Arg180, which we show are involved in ApoE4-CLEAR DNA interactions (Fig. 1c). Therefore, we selected this predicted druggable site within the ApoE4 DNA-binding region to conduct a virtual screen for identifying selective small molecules to inhibit ApoE4::CLEAR DNA interactions.

### High-throughput Virtual Screening predicts potential lead molecules that target the DNA-binding region of ApoE4
To identify such novel small molecules against ApoE4, we performed

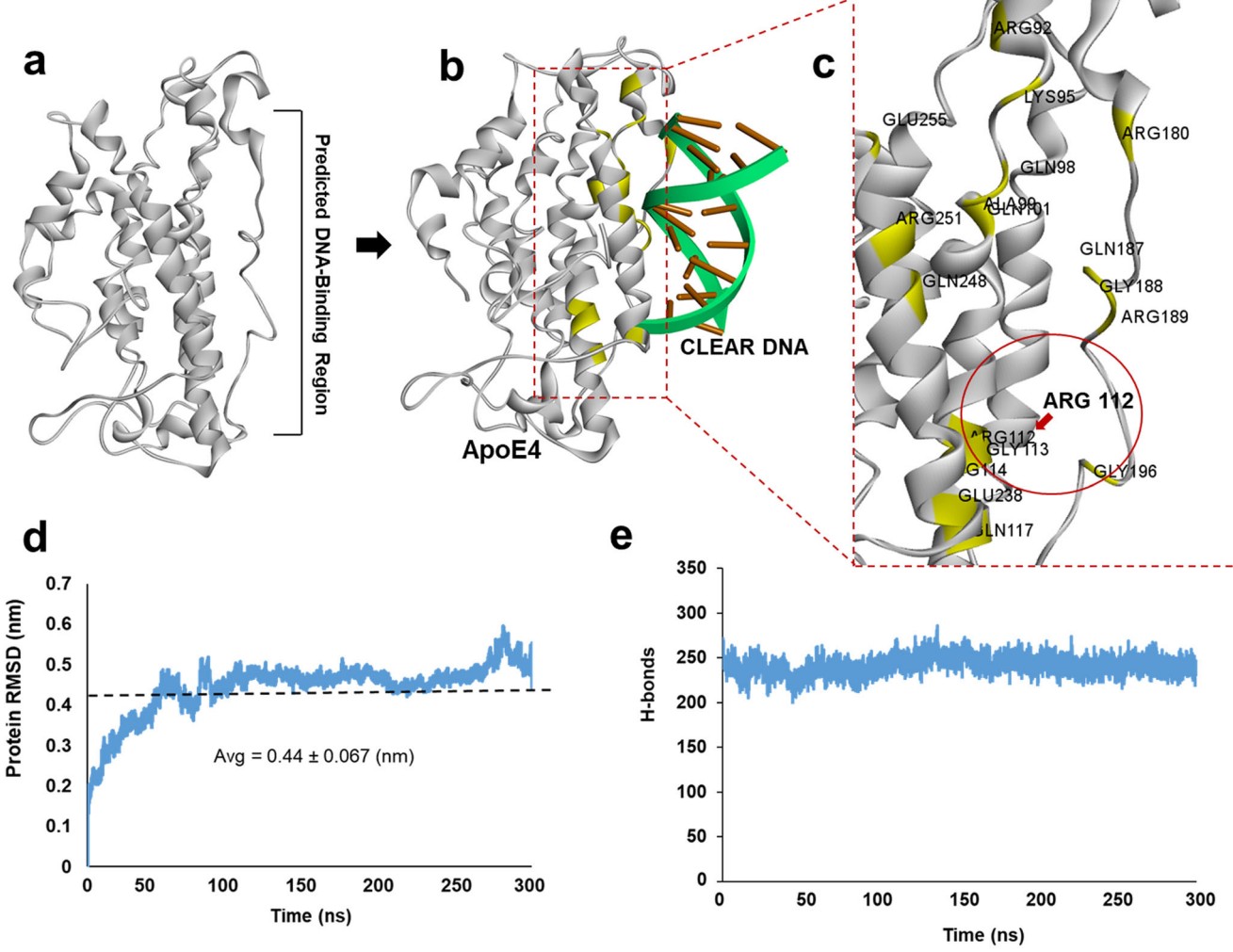

**Fig. 1 Molecular docking predicts a CLEAR DNA-binding region in ApoE4. a** The DNA-binding region of ApoE4, as defined by molecular modeling. **b** The predicted CLEAR-DNA binding pose is shown for ApoE4. **c** Amino acid residues in the region of ApoE4 that interacts with CLEAR DNA are highlighted in yellow. **d** Root Mean Square Deviation (RMSD) of the ApoE4 molecular structure was calculated from 300 ns molecular dynamic simulations, indicating relative conformational stability. **e** Number of internal hydrogen bonds (H-bonds) in ApoE4 protein calculated from 300 ns molecular-dynamic simulation trajectories.

a virtual screening of the ChemBridge small-molecule library (~735,000 molecular structures) for affinity to the predicted druggable site in ApoE4 (Fig. 2b, green mesh). We used the Glide high-throughput virtual screening protocol from the Schrödinger Suite to screen the ChemBridge library (phase I) at low stringency[8–10]. Top molecules from this initial phase were ranked based on predicted docking scores. Compounds having a docking score ≤ −7.3 kcal/mol were then rescreened for docking to ApoE4 under the Schrödinger Glide protocol conducted at high stringency (phase II). Results were then ranked based on the high-precision docking scores from Phase II screening (Fig. 2c). To identify the "best lead compounds," we conducted solvent-based Gibbs binding energy (MMGBSA) calculations within the Schrödinger Prime module (phase III). Results were ranked based on predicted Gibbs binding free energy, selecting the 5 molecules with greatest binding stability (lowest docking free energy) for pursuit (Fig. 2d). Drug binding-pose analysis showed that these small molecules (designated CBA2, 3, 12, 23, and 30) were accommodated within the predicted druggable pocket of ApoE4 (Fig. 2e). The 2D representation of the predicted top molecules shows the structural diversity (Fig. 2f).

**Molecular-Dynamic (MD) simulations of predicted top docking compounds support CBA2 as a potential lead candidate for ApoE4 binding.** Mindful of the importance, over and above docking energies, of the stability and dynamics of a protein-drug complex in identifying best lead compounds in any novel drug-discovery process, we performed individual 200-ns molecular dynamic simulations of ApoE4 complexed with each of the predicted top five molecules. Simulation analyses indicated that among these predicted compounds, only CBA2 had stable dynamics/binding over the entire 200-ns simulation (Supplementary Fig. 1). CBA3 had moderate stability; however, CBAs 12, 23, and 30 had extensive RMSD deviations during the simulations, indicating unstable binding (Supplementary Fig. 1). Drug binding-pose "snapshots" captured from simulation trajectories at multiple time points confirmed that CBA2 binding was stable throughout replicate 200-ns simulations (e.g., see Fig. 3a). As further evidence of ApoE4::CBA2 binding stability, protein/ligand RMSD-ratio analysis showed a stable plateau after 50 ns for ApoE4 complexed with CBA2 (Fig. 3b). Similarly, analysis of the average number of interactions between ApoE4 protein and CBA2 showed stable levels (primarily in the range 4–12)

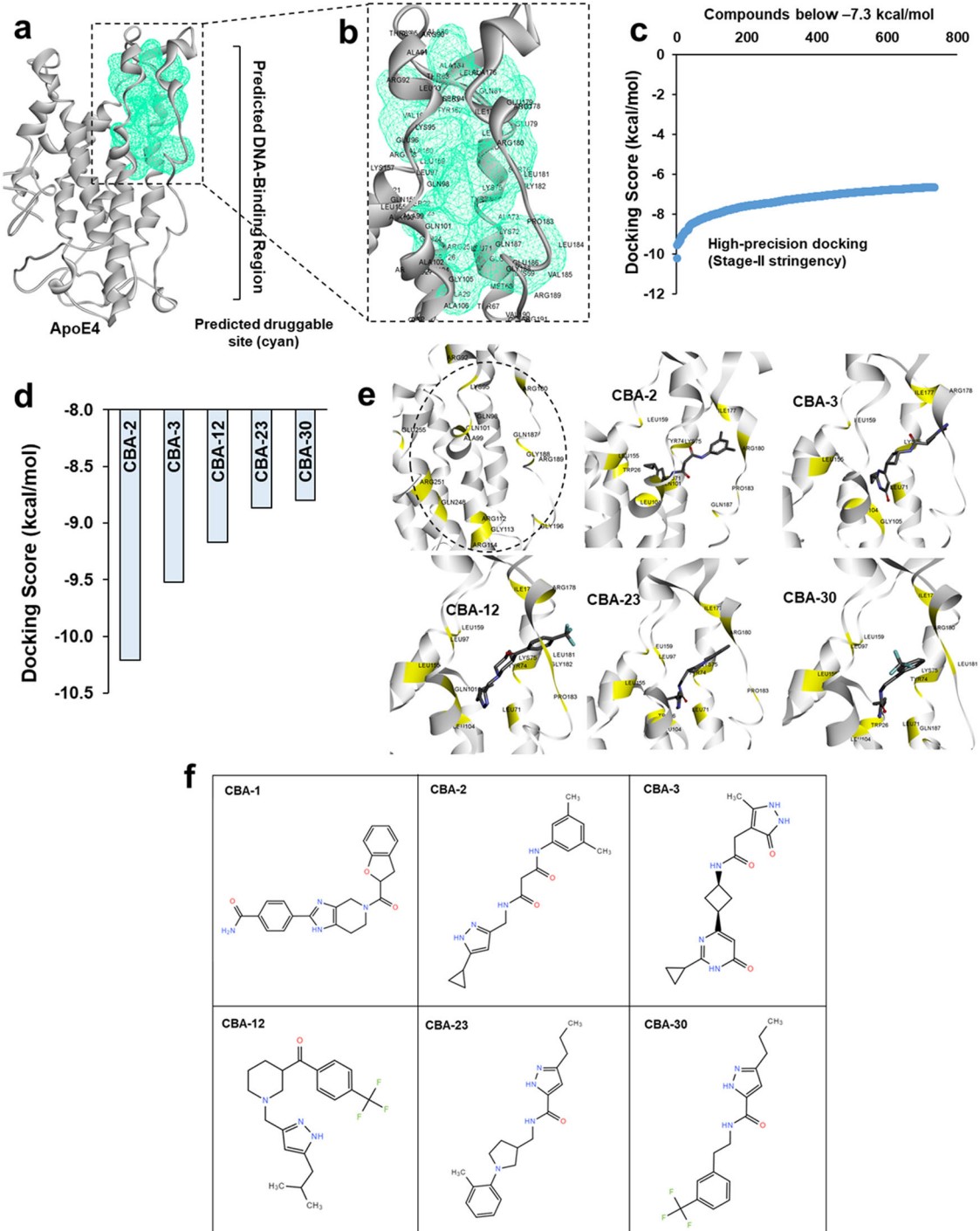

**Fig. 2 Molecular-dynamic simulations predict a druggable pocket (cavity) in ApoE4. a** Receptor-cavity prediction showing a druggable cavity (green) present exclusively in the ApoE4 protein. **b** Magnified view of the predicted druggable cavity in the ApoE4 protein. **c** High-throughput virtual screening predicts docking energies for ChemBridge compounds to ApoE4 protein. Only molecules with binding free energy ΔG< −7.3 kcal/mol are shown. **d** Precision docking energies for the predicted top five compounds. **e** Binding-pose prediction analysis for the predicted top 5 ChemBridge compounds within the ApoE4 DNA-binding site; drug-interacting amino acids are highlighted in yellow. The unoccupied drug-binding pocket is indicated by a dashed oval. **f** Two-dimensional representation of predicted small molecules, having affinity for ApoE4.

indicative of favorable binding (Fig. 3c, d). Analysis of stability of interaction between individual amino acid residues of ApoE4 and CBA2 at the DNA-binding region showed that the majority of interactions between CBA2 and an ApoE4 amino-acid moiety were largely stable throughout the simulation (Fig. 3e). Together, these results predicted that CBA2 comfortably binds at the predicted druggable site within the DNA-binding region of ApoE4.

**CBA2 treatment restores transcription of key autophagy genes in vitro.** We have reported several lines of evidence that ApoE4 protein-CLEAR-DNA interaction impedes the transcription of lysosomal autophagy-related mRNAs—*SQSTM1, LAMP2,* and *MAP1LC3B*—in brain tissue from AD patient carriers of *APOE*ε4,4[4,5]. In order to confirm that the binding of a small molecule, e.g., CBA1, CBA2, or CBA3 to the specific DNA

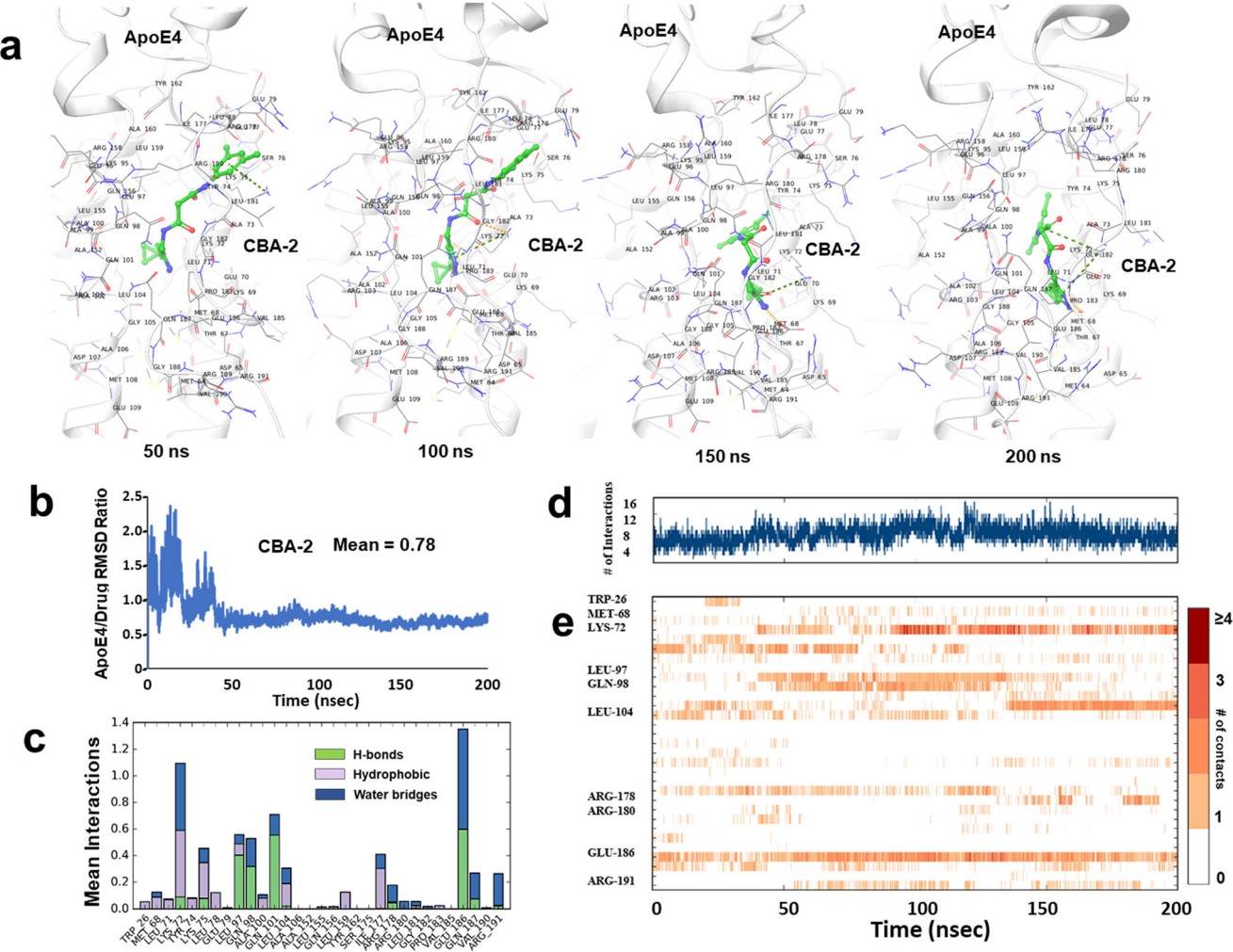

**Fig. 3 Molecular-dynamic simulation of the ApoE4-CBA2 complex. a** Snapshots taken at 50-ns intervals, from a 200-ns MD simulation trajectory of ApoE4-CBA2 complex. **b** Protein/drug ratio of Root Mean Square Deviation (i.e., ligand RMSD) calculated from 200-ns MD simulation trajectories of the ApoE4-CBA2 complex. Simulations were repeated for three times with different seeds for each simulation **c** Type of interactions formed between CBA2 and ApoE4, extracted from simulation trajectories. **d** The total number of interactions between CBA2 and ApoE4 over a 200-ns simulation is indicated. **e** Number/stability of interactions is shown between CBA2 and the amino acid residues of ApoE4 (y-axis) over a 200-ns simulation (x-axis).

binding region on the ApoE4 protein would restore transcription of these same autophagy-associated mRNAs, we turned to primary cultures of astrocytes from targeted-replacement mice (TR) expressing human ApoE3 or -E4 (ApoE-TR3 or ApoE-TR4). By qRT-PCR of mRNA extracted from such mouse primary astrocyte cultures, we found that only CBA2 treatment—and not CBA1, CBA3 (Fig. 4a–c), CBA12, CBA23, or CBA 30 (Supplementary Fig. 2a)—had significant benefit in restoring transcription of autophagy-related mRNAs. The remarkable specificity of CBA2 for ApoE4-expressing cells was clearly demonstrated by treatment of primary astrocytes from ApoE-TR3 mice. Over a CBA2 dose range from 0.5–50 μM, mRNA levels for *Sqstm1, Lamp1, Map1lc3a,* and *Map1lc3b* show a clear concentration-dependent response in ApoE4-TR, rising 2.4- to 2.8- fold at the highest concentration, while astrocytes from ApoE3-TR mice were unresponsive to CBA2 treatment (Fig. 4d).

We also used Western-blot analysis to assess protein levels of SQSTM-1/p62, LC3B, and LAMP2 in T98G cells overexpressing either ApoE3 or ApoE4, with and without CBA2 treatment. As expected from the increase in mRNA transcript levels, the associated protein levels were also increased in T98G-E4 but not in T98G-E3 cells (Fig. 4e, f, and Supplementary Fig. 3a).

Furthermore, to confirm that treatment of CBA2 restores autophagy itself we measured autophagy in T98G cells expressing either ApoE3 or ApoE4, treated with rapamycin, an autophagy inducer. Rapamycin induced autophagy in T98G-E3 cells; however, it did not induce autophagy in T98G-E4 cells (Fig. 4g). Treatment with CBA2 significantly restored autophagy in T98G-E4 cells, substantiating our transcriptional level analysis (Fig. 4g, h). Notably, CBA2 treatment did not have any effects in T98G-E3 cells (Supplementary Fig. 3b, c). These findings are consistent with the tendency of ApoE4 to suppress expression of autophagy-related genes and the ability of CBA2 to restore their expression.

**CBA2 treatment protects against amyloid-like (Aβ₄₂) aggregation and restores chemotaxis behavior in a *C. elegans* AD model**. We previously showed that expression of human Aβ$_{1-42}$ in the neurons of *C. elegans* induces amyloid-like deposits[11,12] that are associated with a decline in chemotaxis toward *n*-butanol, a potent chemoattractant[13,14]. Our nematode model recapitulates the AD-like scenario in that co-expression of human ApoE4 and Aβ$_{1-42}$ in *C. elegans* (*ApoE4; Aβ::mcherry* worms) results in a 45% increase in Aβ::mcherry aggregation relative to *ApoE3; Aβ::mcherry* worms (Fig. 5b). We anticipated

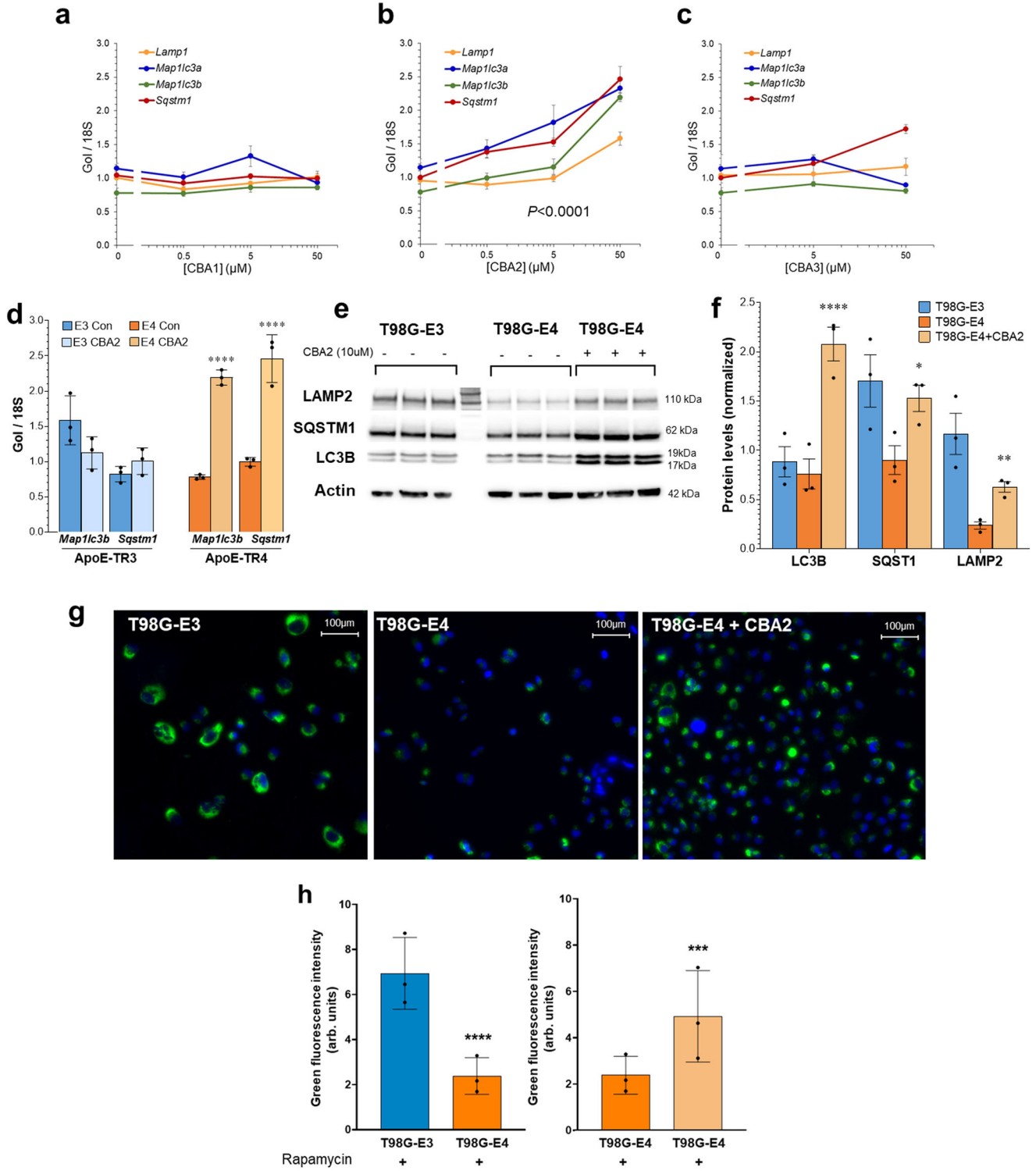

that treatment of these worms with CBA2 would improve their lysosomal autophagy and thus protect against Aβ aggregation (Fig. 5a, b). This proved to be case when Aβ::mcherry worms co-expressing either human ApoE4 or ApoE3 were treated from hatch for 7 days with 10-μM CBA2 or vehicle and then assessed for Aβ-mCherry aggregates as quantified from fluorescence images (Fig. 5a, b). CBA2 treatment significantly reduced Aβ$_{42}$::mCherry aggregates in worms expressing ApoE4, bringing them to the level seen in ApoE3-expressing worms. This finding parallels our earlier observation that AD patients carrying

$APOE$ε4,ε4 have more Aβ aggregates than $APOE$ε3,ε3 patients. Further, lipid-vesicle transduction[15] of ApoE4 protein into worms expressing Aβ$_{1-42}$ elicited a significant decline in che-motaxis relative to worms receiving ApoE3, an event that was reversed by exposure to 10-μM CBA2 (Fig. 5c).

When we assessed the ability of CBA2 to restore normal levels of key autophagy proteins by treating Aβ$_{1-42}$-expressing trans-genic C.elegans for 5 days with 10-μM CBA2 or vehicle, Western-blot analysis showed that the protein levels of SQST-1, LGG-2, and LAMP-2 were significantly increased after CBA2 treatment

**Fig. 4 CBA2 treatment restores transcription of genes suppressed by ApoE4. a–c** Primary astrocytes from ApoE-TR4 mice were exposed to low-glucose medium, and CBA1, −2, or −3 was applied at the indicated concentrations. After 20 h, mRNA levels of *Lamp1, Map1lc3a, Map1lc3b,* and *Sqstm1* were determined by qRT-PCR. ****$P < 0.0001$ for main effect of CBA2. **d** CBA2 (50-μM) was applied to primary astrocytes from ApoE-TR3 or ApoE-TR4 mice for 20 h in low-glucose medium, and mRNA levels of *Map1lc3b* and *Sqstm1* were determined by qRT-PCR. ****$P < 0.0001$ *versus* control, ($N = 3$ repeats, represented as individual data points). **e** Western-blot analysis of key autophagy proteins from T98G-E3 or T98G-E4 cells ± CBA2. Protein bands shown in (**e**) were derived from the same blot that was stripped and re-probed with different antibodies. **f** Histogram shows normalized band intensities from Western blots of T98G-E4 cells ± CBA2 (10 μM). Exposure to CBA2 significantly increased protein levels in T98G-E4 cells; Values are mean ± SEM; significance between CBA2-treated cells and control cells was determined by 2-way ANOVA and the Bonferroni *post hoc* test, ($N = 3$ repeats, represented as individual data points). **g** Images of T98G-E3 or E4 cells treated with rapamycin, an autophagy inducer ± CBA2 treatment. Intensity of green fluorescence is directly proportional to the autophagy induction. **h** Histogram of green fluorescence intensity per cell calculated from fluorescent images, error bars represent SEM. ***$P < 0.004$, and ****$P < 0.0005$, via 1-tailed *t* test for an *N* of 3 biological repeats, represented as individual data points, each with 15 to 20 images per group per experiment.

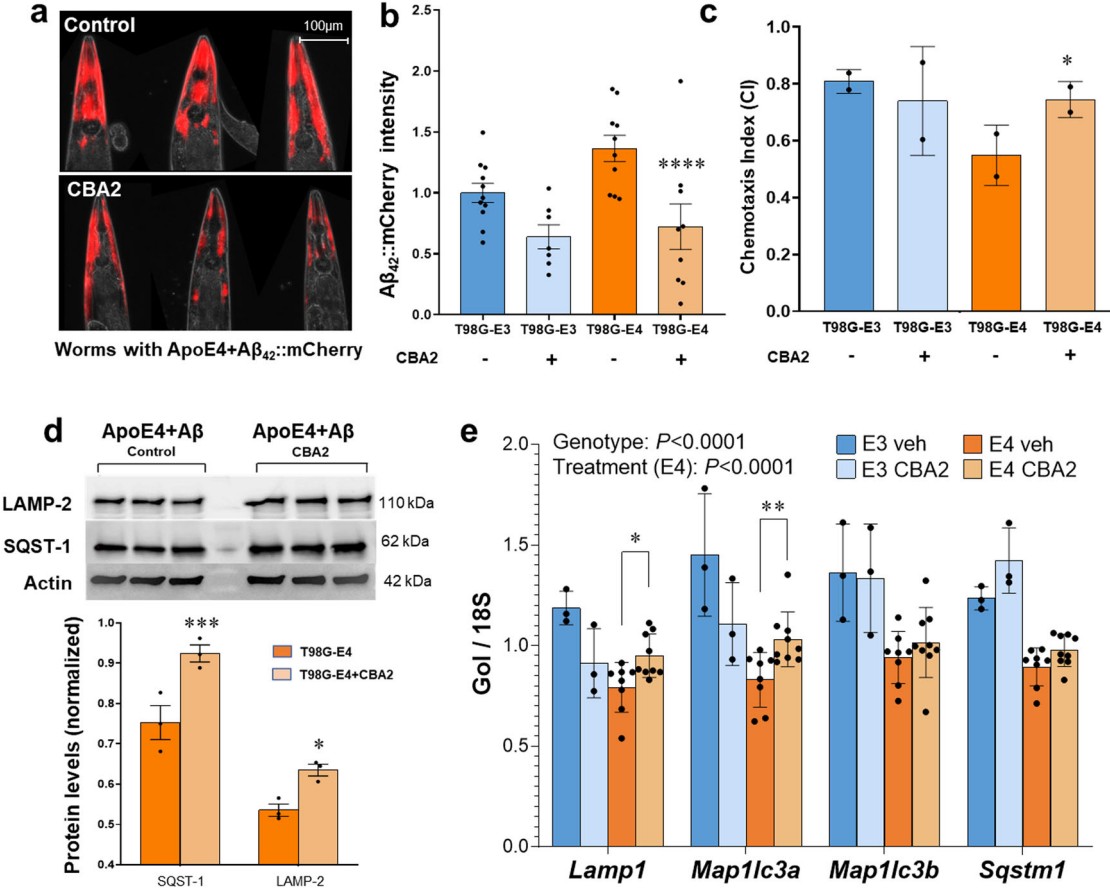

**Fig. 5 CBA2 treatment reverses the effects of ApoE4 in a *C. elegans* model of AD and in targeted-replacement mice. a** Images of transgenic *C. elegans* expressing Aβ$_{42}$::mcherry in neurons. Red fluorescent aggregates are diminished after exposure to 10-μM CBA2 (lower panel). **b** Histogram of mean normalized Aβ$_{42}$::mCherry intensities calculated from fluorescence images. Significance of inter-group differences was determined by 2-way ANOVA and Bonferroni *post hoc*; ****$P < 0.0001$ for $N = 15$–20 worms/group, represented as individual data points. **c** Chemotaxis index (CI) assessed on worms transduced with ApoE3 or ApoE4 protein, and treated with CBA2 for 5 days. CBA2 treatment improved chemotaxis in worms relative to either untreated worms or those receiving ApoE3 protein; *$P < 0.02$ by CHI.SQ test for $N = 40$–100 worms/group. Values are mean ± STDEV, ($N = 2$ repeats, represented as individual data points). Significance between CBA2-treated worms and control worms was determined by CHI.SQ test. **d** Western-blot analysis of key autophagy proteins in worms expressing ApoE4 ± CBA2. Protein bands shown in (**d**) were derived from the same blot that was stripped and re-probed with different antibodies. Histogram shows normalized band intensities calculated from western-blot images. Exposure to CBA2 significantly increased levels of SQST-1 and LAMP-2 in C.elegans. Significance of inter-group differences was determined by 2-way ANOVA and Bonferroni *post hoc*; *$P < 0.02$, ***$P < 0.0003$ *versus* control. **e** ApoE-TR3 ($N = 3$) and -TR4 mice ($N = 8$–9) were treated 12 h with CBA2 (47 μg/kg) or vehicle. RNA was prepared from liver, and mRNA levels of *Sqstm1, Map1lc3a, Map1lc3b,* and *Lamp1* were assessed by qRT-PCR. *$P = 0.0492$; **$P = 0.0066$; main effect of CBA2 in ApoE-TR4 mice: $P < 0.0001$. Error bars indicate SEM. Three-way ANOVA was applied to test for potential main effects and interactions between the three variables in the overall experiment: genotype, treatment, and gene of interest. Differences within individual genes of interest were determined by 2-way ANOVA and Bonferroni *post hoc*.

(Fig. 5d). In contrast, CBA2 did not elicit any significant change in levels of these autophagy proteins in *C. elegans* expressing ApoE3, thus confirming the ApoE allelic specificity of CBA2 for ApoE4 (Supplementary Fig. 2b, c).

**CBA2 treatment of ApoE4-expressing mice reverses suppression of autophagy-related genes.** To test the potential for an ApoE4-binding compound to pharmacologically reverse the genetic effects of ApoE, we administered CBA2 to mice expressing human ApoE3 or ApoE4 via targeted replacement (TR). An initial toxicology screening of CBA2 revealed no elevation of markers of renal and hepatic stress 24 h after i.p. administration of CBA2 at doses up to 100 µg/kg (Suppl. Table 1). Computed blood-brain barrier permeability values (logP) for CBA2 suggest that it would not partition to the CNS efficiently. Therefore, we focused our initial tests for functional effects of CBA2 on liver, another organ with significant levels of ApoE expression. ApoE-TR3 and ApoE-TR4 mice were fasted to induce expression of MiT/TFE family transcription factors. CBA2 (47 µg/kg) or vehicle was then administered i.p., and mRNA levels for four autophagy-related genes were assayed by qRT-PCR 12 h later. ApoE-TR3 mice showed higher expression of *Lamp1, Map1lc3a, Map1lc3b,* and *Sqstm1* than that detected in vehicle-treated ApoE-TR4 mice; main effect of gene: $P < 0.0001$ (Fig. 5e). ApoE-TR4 mice responded to CBA2 with an increase in the expression of these genes; main effect of drug: $P < 0.0001$. However, ApoE-TR3 mice did not; interaction between genotype and drug: $P < 0.034$. A *post hoc* test revealed a significant elevation of *Lamp1* and *Map1lc3a* in CBA2-treated ApoE-TR4 mice.

## Discussion

Autophagy is a critical pathway in protein homeostasis, and its obstruction is related to aggregate accumulation in diverse model systems[16–18]. In addition to our studies of human tissue from AD-*APOE*ε3,3 (AD 3,3) and AD-*APOE*ε4,4 (AD 4,4) patients, inheritance of *APOE*ε4 has been shown in several studies to obstruct autophagy. For example, Simonovitch et al.[3] showed that autophagic flux is lower in ApoE4-expressing astrocytes than in those expressing ApoE3. Moreover, Cataldo et al. (2000) showed that early endosomes, which are part of an autophagy-related pathway, are enlarged in brains of early-stage AD patients carrying an *APOE*ε4 allele[19]. Perhaps in association with this, Aβ is trafficked through the endosomal system to the lysosome where lysosomal clearance of Aβ is deficient in the presence of ApoE4, but not ApoE3[20]. Badia et al.[21] showed that even young carriers of *APOE*ε4 have reduced lymphocytic levels of TFEB, the transcription factor for transcription of mRNAs for production of necessary proteins for lysosomal autophagy. Such limitation of the production of these proteins is accompanied by elevation of both Aβ as well as hyperphosphorylation of tau[22], thus corroborating the importance of activity of MiT/TFE family transcription factors for efficient lysosomal autophagy and clearance of large entities such as aggregates and spent organelles.

The research presented here establishes a proof-of-concept for restoring normal autophagy with drugs that specifically target ApoE4. Our establishment of a direct relationship between inheritance of *APOE*ε4 alleles and a dramatic reduction in transcript levels for three mRNAs that encode proteins required for lysosomal autophagy in brain tissue from AD carriers of *APOE*ε4,4 but not *APOE*ε3,3[4] suggests a need for a targeted drug to counteract the effect of ApoE4 on lysosomal autophagy. We also established, by both computer modeling and by several biological assays, that ApoE4 binds to CLEAR motifs but ApoE3 and ApoE2 do not[4]. Importantly, we demonstrate here that small molecules with affinity for the DNA-binding region of ApoE4 can block ApoE4-mediated inhibition of lysosomal autophagy in both in vitro and in vivo models of AD-like aggregation. Moreover, successful interference with such binding alleviates Aβ aggregation[3,23,24], thus supporting our previous work identifying autophagy as a crucial aggregate-clearance control mechanism with therapeutic potential to limit accumulation of Aβ plaque and tau tangles in AD. We believe that this ApoE4-related effect is generalizable to other neurodegenerative diseases characterized by aggregate accumulation arising from failure of lysosomal autophagy[4,16,25,26]. We note here that autophagy-gene transcript levels indicate a much larger protective effect of CBA2 than that of the corresponding protein levels. This is to be expected since mRNA transcripts are directly impacted by ApoE4 and rescued by CBA2 disruption of ApoE4 competition with members of the MiT/TFE family, whereas protein levels instead reflect the cumulative total product of translation, which would require a much longer drug exposure to be similarly affected. Moreover, the process of active autophagy degrades SQSTM-1/p62 and several other proteins key to the process[27].

Our strategy for drug discovery began by identifying a novel druggable site in ApoE4, which we then used to screen structural drug libraries for binding avidity. To confirm selective activity in the presence of ApoE4 but not ApoE3, lead candidates were tested in multiple cell-culture and intact-nematode models that express human ApoE variants or which were transfected with exogenous protein. In multiple bioassays, CBA2—a novel hit-to-lead compound—emerged as the top candidate with respect to rescue of aggregation-mediated phenotypes with negligible toxicity across a wide range of concentrations and, importantly, with specificity for ApoE4. Transcript-level rescue of autophagy genes suppressed by ApoE4 implies that CBA2 acts to impede ApoE4 binding to CLEAR DNA. Furthermore, measurement of autophagy in live cells showed that active autophagy levels were low in T98G-E4 cells compared to T98G-E3 cells. This observation substantiates our hypothesis that the presence of ApoE4 thwarts lysosomal autophagy. The fact that CBA2 treatment significantly improved autophagy induction in T98G-E4 cells signifies the therapeutic value of targeting ApoE4. Treatment of CBA2 protected against Aβ::mCherry aggregation in worms expressing human ApoE4 protein, suggesting the importance of rescuing autophagy in protection against amyloid aggregation. It is important to note that the reduction in aggregates (though not significant), seen in worms expressing ApoE3 could be due to marginal off-target effects of small molecule, CBA2, especially in C. elegans environment. Further the potential of CBA2 to facilitate autophagy provides a personalized-medicine approach, which for the first time shows the possibility for a targeted intervention specifically for individuals inheriting *APOE*ε4 gene(s).

It is particularly encouraging to observe the expected effect of CBA2 on expression of autophagy-related genes in ApoE-TR4 mice. CBA2 as a hit-to-lead compound was well tolerated in this mammalian model and showed promising efficacy in liver. Eventually, it will be important to test for the activity of an ApoE4-binding compound in the brains of mice in an AD-relevant model, as the effects of ApoE4 may only be manifest under conditions in which lysosomal autophagy is induced by a stress reaction such as that engendered by Aβ accumulation. Notably, our original analysis of human brain samples found that *APOE*ε3,3 individuals showed an elevation of autophagy-related genes only when the pathology of AD prevailed[4].

Together, our data substantiate the premise that a small molecule can be identified, which specifically targets ApoE4, while having no interaction with ApoE3. This suggests that a drug can be developed that will be free from complications in *APOE*ε3,4 heterozygous individuals due to lack of interaction

**Table 1 qRT-PCR primers.**

| Gene (direction) | sequence | species |
| --- | --- | --- |
| Lamp1 (F) | 5′-AGT GGG AGT TGC GGT ATC AA-3′ | M. musculus |
| Lamp1 (R) | 5′-GGC TAG AGC TGG CAT TCA TC-3′ | M. musculus |
| Map1lc3a (F) | 5′-CGC TAC AAG GGT GAG AAG CA-3′ | M. musculus |
| Map1lc3a (R) | 5′-GCG GCG CCG GAT GAT-3′ | M. musculus |
| Map1lc3b (F) | 5′-CAA GAT CCC AGT GAT TAT AGA GCG A-3′ | M. musculus |
| Map1lc3b (R) | 5′-TGC AAG CGC CGT CTG ATT AT-3′ | M. musculus |
| Sqstm1 (F) | 5′-AGG AAC AGA TGG AGT CGG GA-3′ | M. musculus |
| Sqstm1 (R) | 5′-CCG GGG ATC AGC CTC TGT AG-3′ | M. musculus |

with ApoE3. Based on these results, we propose CBA2 as a lead molecule to forestall negative traits in those inheriting *APOE*4 allele(s). Nevertheless, detection of effects of inhibiting ApoE4 DNA binding may await identification of compounds that have more promising logP values than that exhibited by CBA2.

We reckon that our novel drug-discovery strategy, which allowed us to: i) identify a novel druggable site in ApoE4; ii) successfully target that site with one of our novel lead compounds, e.g., CBA2; and to iii) preclude ApoE4 binding to CLEAR DNA, has the potential to be developed as a personalized-preventive medicine approach, i.e., a therapeutic intervention for individuals inheriting one or both alleles of *APOE*ε4. Further, as we focus here on a remedy for the 12–15 times increase in risk for development of AD in inheritors of *APOE*ε4 from both parents[1], it is possible that transformation of ApoE4-related crises via interventions such as offered here will be useful for proper lysosomal autophagic functions not only in brain cells but also in cells of other organs.

The drug-discovery approach taken here, and the small molecule(s) it has identified, may for the first time provide an effective preventative approach to directly foil the devastating effects of inheriting *APOE*ε4 alleles, not only for development of Alzheimer's disease but also with respect to the many other maladies associated with the inheritance of the *APOE*4 allele(s)[1,28,29].

## Methods

**Molecular Dynamic simulation and druggable-pocket prediction**. Three-dimensional structures of ApoE3 and ApoE4, as previously modelled and reported[4], were used for molecular-dynamic (MD) simulation studies. MD simulations were performed in the Desmond simulation package[7,9,30]. The target protein structure was immersed in the triclinic box, containing Simple Point Charge (SPC) water and NaCl as counter-ions. To mimic physiological conditions, NaCl was further added to 0.15 M in the simulation system. The entire model system was energy minimized for 5000 steps before equilibration. Using the NVT/NPT method, the entire simulation system was equilibrated for 300 ps, after which the MD run was performed for 200 ns, and trajectories were saved every 100 ps. Trajectories were analyzed using the Simulation Interaction Diagram protocol from Maestro/Schrodinger. The simulated ApoE4 structure was then used to predict likely drug-binding cavities using BIOVIA Discovery Studio's Receptor Cavities plug-in.

**High-throughput virtual screening of the ChemBridge structural library**. To identify the best candidates in the ChemBridge library that target ApoE4, we used the Glide docking protocol[7,8,10]. Briefly, the pocket identified as the DNA-binding region of ApoE4 was placed within a docking grid box. The full ChemBridge structural library of ~780,000 compounds were prepared using the Ligand Preparation Wizard from the Schrodinger Small Molecule Drug Discovery Suite. Prepared ligands were initially screened under the HTVS protocol in the Glide docking package. For high-stringency (phase II) docking, we employed high-precision mode within the Glide docking package. Highest-stringency (phase III) screening used the Molecular Mechanics Generalized Born Solvent Accessible (MM-GBSA) protocol from Schrodinger Prime module. All three-dimensional results were analyzed in Maestro and BIOVIA Discovery Studio visualizers.

**Tissue culture and maintenance**. Stable transformants of human glioblastoma (T98G) cells were generated as described previously by Wang et al.[31]. T98G cells, carrying transgenes expressing either ApoE3 or ApoE4, were grown in Dulbecco's Modified Eagle's medium (DMEM) (Cat. No: 119950065, Thermo Fisher Scientific, Waltham, MA, USA) which was supplemented to 10% (v/v) fetal bovine serum (FBS) (Cat. No: 16000044, Thermo Fisher Scientific). Primary astrocytes were cultured from neonatal mice as described previously[32]. Primary astrocytes were cultured from neonatal mice as described previously (Sung et al., 2023). These cells were subjected to low serum (0.5%) and glucose (0.2 mM) concentrations during drug treatments to favor activation of TFEB.

**Antibodies and reagents**. The following commercially available antibodies were used: anti-SQSTM-1/p62 (BD610832, BD Biosciences), anti-actin (D6A8 from Cell signaling Technologies), anti-LC3B (NB600–1384, Novus Biologicals), and rabbit monoclonal anti-LAMP-2 (NBP2-67298; Novus-Biologicals).

**RT-PCR amplification**. Total RNA was extracted from cells using the Qiagen RNA extraction kit (RNeasy Plus Mini kit #74134), according to the manufacturer's instructions. Quality and quantity of the extracted RNA was determined by Agilent bioanalyzer. RT reactions were performed on equal amounts of RNA using single-step RT-PCR reagents. Primers were synthesized by IDT (sequences provided in Table 1). PCR amplification was run for 40 cycles comprising 95 °C for 15 s, followed by 60 °C for 60 s, and 40 °C for 60 s. Values were obtained by interpolation in a standard curve generated from a pool of all samples, and the quantity of the gene of interest was normalized to that of 18 S rRNA.

**Western blots**. Human T98G cells were maintained in cell culture medium (DMEM; Invitrogen/Life Technologies, Grand Island, NY, USA) supplemented with 10% v/v fetal bovine serum (FBS). Protein from cell lysates was quantified with Bradford reagent (Bio-Rad; Hercules, CA, USA), and 30 μg protein aliquots were electrophoresed for 2 h at 100 V on 4–20% gradient bis-tris acrylamide gels (BioRad Life Science, Hercules, CA, USA) and transferred to PVDF membranes. Membranes were blocked with BSA blocker (Pierce) and incubated overnight at 4 °C with

primary antibodies to SQSTM-1/p62, LAMP2, LC3B, or GAPDH. After five washes of 5 min each, membranes were incubated 1 h at ~20 °C with HRP-conjugated secondary antibody—either goat anti-rabbit IgG or goat anti-mouse IgG (both from Cell Signaling Technologies, Danvers, MA, USA)—used at 1:3000 dilution, and developed using an ECL chemiluminescence detection kit (Pierce). For each experimental repeats, blots were striped and re-probed with different primary antibodies. Data were digitized and analyzed using ImageJ software (NIH).

**C.elegans strains**. *C. elegans* strains were cultured using standard methods as previously described[33]. The strains used in this work include CL2355 (obtained from the Caenorhabditis Genetics Center); UA353 (baIn51[$P_{eat-4}$::APOEε3, $P_{unc-54}$::tdTomato]; baIn34[$P_{eat-4}$::Aβ, $P_{myo-2}$:: mCherry]; adIs1240[$P_{eat-4}$::GFP]), and UA355 (baIn51[$P_{eat-4}$::APOEε4, $P_{unc-54}$::tdTomato]; baIn34[$P_{eat-4}$::Aβ, $P_{myo-2}$:: mCherry]; adIs1240[$P_{eat-4}$::GFP]); and adIs1240 [$P_{eat-4}$::GFP]. All strains except CL2355 were generated by and are a kind gift from Guy Caldwell.

**Chemotaxis**. Chemotaxis was assessed in *C. elegans* strain CL2355, with pan-neuronal expression of Aβ$_{1-42}$. ApoE3 or ApoE4 protein (75 μM) was introduced by lipid-vesicle trans-duction using PULSin (Polyplus) reagent[15]. Synchronized eggs were fed from hatch ± CBA2, and maintained at 20 °C for 7 days. At 7 days post-hatch, worms were washed in S-buffer to remove bacteria and other debris. Approximately 50–100 worms per assay were placed at centers of 100-mm culture plates spotted at one edge with ~5 μL n-butanol as chemo-attractant plus ~5 μL sodium azide to immobilize attracted worms, and ~5 μL S-buffer plus ~5 μL azide at the opposite edge as a control. Assay plates were held at 20 °C, and chemotaxis was scored every 30 min. The 'Chemotaxis Index' (CI) was calculated as in previous studies[11] as CI = [(worm# near attractant) − (worm# near control)]/(total worms/plate).

**Fluorescence imaging**. The intensity of Aβ::mCherry aggregates was quantified after imaging (Keyence fluorescence microscope, USA) of worms at 20× magnification. For quantitation, images of 10 to 15 worms were analyzed using FIJI ImageJ image; resulting data were analyzed and plotted within Microsoft Excel.

**Mouse pilot treatment**. Male mice (5 weeks of age) homozygous for ApoE3-TR or ApoE4-TR loci[34] were fasted for 9 h, then CBA2 was injected i.p. at 47 μg/kg; controls received an equal volume of vehicle (0.5% DMSO in saline). After another 12 h, mice were euthanized and total RNA was prepared from tissues by RNeasy Plus (Qiagen). All procedures were approved by the Central Arkansas Veterans Healthcare System Animal Care and Use Committee.

**Autophagy detection assay**. Autophagy induction was deter-mined using the autophagy detecting kit (Abcam, ab139484), following the manufacturer's protocol. In brief, T98G-E or T98G-E4 cells were plated in 8-chamber slides at ~15,000 cells per well. Cells were treated with rapamycin (1 μM final concentration) ± CBA2. Before staining, cells were washed twice with 1X assay buffer with 5% FBS. Cells were stained with microscopy dual detection provided in the kit and analyzed by fluorescent microscopy (Keyence BZ-X810 microscope). Fluorescent images were quantified using FIJI ImageJ software.

**Statistical analysis**. For qRT-PCR, western blot experiments, and in vivo administration of CBA2, significance of inter-group dif-ferences was determined by 2-way ANOVA and Bonferroni *post hoc* analysis. For the in vivo experiment 3-way ANOVA was also utilized to test potential main effects and interactions between the three variables: genotype, treatment, and gene of interest. ANO-VAs and *post hoc* tests were performed with GraphPad Prism 9.5.1. For chemotaxis experiment, significance was determined based on CHI.SQ test. For autophagy detection assay, significance was determined using heteroscedastic *t* test.

**Reporting summary**. Further information on research design is available in the Nature Portfolio Reporting Summary linked to this article.

## Data availability

Initial coordinates, simulation input files, and coordinate files associated with in silico analyses are available from the corresponding author upon request. No other data associated with this study comprise large sets that would be valuable for data mining or manipulation. Uncropped Western blots presented in the paper can be found in the supplementary section as Supplementary Fig. 4. The source files for all the graphs presented in the paper can be found in the supplementary section, and also available in Mendeley Data[35]

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

## Acknowledgements

This work was supported by funding from the following grants: Program Project grant 2P01AG012411-17A1 (W.S.T. Griffin, P.I.); R01AG084472 (W.S.T. Griffin, P.I.); R01AG071782 (S.W.B, P.I.); and R01AG084473 (S.W.B, P.I.) from the National Institute on Aging (NIA/NIH); Merit Review Award I01 BX001655 and Senior Research Career Scientist Award IK6 BX004851 to R.J.S.R. from the U.S. Dept. of Veteran Affairs; and an award to M.B. from the Inglewood Scholars Program. Further funding was provided by the Windgate Foundation, the Philip R. Jonsson Foundation, the Ottenheimer Brothers Foundation, and the STOP Alzheimer's campaign at the UAMS Foundation.

## Author contributions

M.B., and W.S.T.G. conceived the idea. M.B. performed computational modeling, High-throughput Virtual Screening, and molecular dynamic simulations with contributions from A.G. J.N., and L.L. performed cell culture experiments, including RT-PCR and Western blot analysis. H.A. performed C.elegans imaging with guidance from S.A., and R.A. G.C. generated C.elegans strains expressing human ApoE protein. S.W.B. carried out in vivo drug treatment and analysis in the mouse models. M.B. and W.S.T.G. wrote the manuscript, while S.A., S.W.B., and R.J.S.R. reviewed the data and edited the manuscript. All authors have given their approval to the final version of the manuscript.

## Competing interests

The authors declare the following financial interests/personal relationships which may be considered as potential competing interests: The University of Arkansas for Medical Sciences (UAMS) holds patents on the molecules described in this study. A potential royalty stream to M.B., S.W.B, S.A, R.J.S.R., and W.S.T.G. may occur consistent with UAMS policy. All other authors declare no competing interests.
