## [Peer Review File · Communications Biology]

Reviewers' comments:

Reviewer #1 (Remarks to the Author):

The authors identified a novel druggable site in ApoE4 and its potential binding sites with CLEAR-DNA motif. High-throughput virtual screening of structural drug libraries for binding avidity further pinpointed CBA2 as a potential lead candidate. The finding is impactful as ApoE4 protein-CLEAR-DNA interaction impedes lysosomal autophagy function. Additionally, the authors showed that CBA2 treatment can reverse ApoE4-mediated inhibition of lysosomal autophagy in both mouse and *C.elegans* models, in turn clearing AD-like aggregates. This paper is well-written and easy to follow. The work is interesting to the broader field.

Specific recommendations:

1. Microglia also expresses APOE. Please rationalize using only astrocyte culture but not microglia in the in vitro model.
2. Has the expression of autophagy proteins in the primary astrocyte cultures been measured via Western blots? If so, what was the outcome?
3. Possible to intracranially inject CBA2 into ApoE-TR4 vs ApoE-TR3 mice to validate its effect in the brain?
4. Were T98G-E3 cells also treated with CBA2? Fig. 4e only had data of T98G-E4 treated with CBA2.
5. In Fig. 5b, why would E3+CBA2 also have a reduction, though not significant, still a 1/3 reduction? Seeing a reduction in the APOE3-expressing worms upon CBA2 treatment weakens the argument that CBA2 specifically blocks APOE4-induced A β aggregation in the *C.elegans* model.
6. On page 11, the statement "while *Sqstm1* was not among the individual genes significantly elevated in vivo, one of this gene's transcriptional starts lacks any obvious CLEAR site; an alternative start manifests two candidates within 1 kb, and it is tempting to speculate that this may be the promoter that responded in cultured astrocytes" is difficult to follow. Which piece of data was this statement referring to?
7. Please consolidate all the statistical analyses into a single statistical analysis section located in the methods section. As currently written, information on statistical analyses appears variably in the figure legends, associated with specific analytical procedures, and briefly in the results section. Plus, the statistical analysis for your Fig.5e is hard to understand. Why was 3-way ANOVA performed?

General suggestions:

1. Throughout the manuscript, different terms "APOE ϵ 3,3", "APOE ϵ 3, ϵ 3", "APOE ϵ 4,4", "APOE ϵ 4, ϵ 4" and so on were used when referring to APOE genotypes. The nomenclatures are inconsistent.
2. Please re-check the entire manuscript for typographical and formatting errors. E.g., on page 3 in the intro, "... AD-associated hallmark pathognomic aggregates of AD...". On page 8, "*sqst-1*, *lgg-2*, and *Imp-2* should be in uppercase"
3. There are areas where words need to be added or corrected. For example, on page 7, there is a missing labeling of Fig in "... we found that only CBA2 treatment, not CBA1 or CBA3 (Fig a & c)...". On page 8, there is a wrong labeling of Fig in "... with CBA2 would improve their lysosomal autophagy and thus protect against A β aggregation (Fig 4g)".
4. Define ApoE-TR3 mice when it first appeared in the main text.
5. Figure legends:
 - a. Specify your sample size throughout the different experiment
 - b. Fig. 1c, indicates what yellow regions are. Are they the residues that directly interact with CLEAR-DNA?
 - c. Fig. 3c, specifies the color-coded interactions. What is the purple, blue, and green region?

Reviewer #2 (Remarks to the Author):

APOE4 is the strongest risk of late-onset AD and is associated with the autophagic/lysosomal failure in AD. In this manuscript, Balasubramaniam et al. employed a computational approach to identify small molecules that impede ApoE4::DNA binding. The leading compound CBA2 restored the expression of several autophagy genes in both in vitro and in vivo models and protected against A β 42 aggregation in *C. elegans* AD model. Overall, this study is novel and interesting, providing a new strategy to regulate autophagic/lysosomal function in AD.

Comments:

1. The structures of CBA1, 2, 3, 12, 23 and 30 should be presented in Fig. 2d. Fig. 2e shows 5 compounds, but the exact structures are not clear.
2. The names of autophagy proteins are not consistent between Supplementary Figure 2b and Supplementary Figure 2c.
3. Numbers represent the migration of molecular weight markers (in kilodaltons) are missed in all western blot images.
4. CBA2 restored several autophagy genes in both in vitro and in vivo models. To strengthen the overall conclusion of this paper, the authors could perform autophagy assays to directly monitor autophagy in living cells after CBA2 treatment.

We thank the reviewers for their positive comments toward improving the quality of the manuscript. As per the suggestions, we have made substantial modifications to the manuscript, including the new data on measuring autophagy in live cells, treated with or without CBA2. Responses to the individual comments are as follows:

Reviewer #1

Specific recommendations:

1. Microglia also expresses APOE. Please rationalize using only astrocyte culture but not microglia in the in vitro model.

We agree with the reviewer that determination of the effects of ApoE4 on proteostasis in microglia will be essential for providing a full view of ApoE biology. However, that was not the objective of this paper. Here, we sought merely to model the mechanism by which ApoE4 impacts gene expression in order to test the effects of drug-like molecules on this process. To this end, astrocytic cells provided a useful model system.

2. Has the expression of autophagy proteins in the primary astrocyte cultures been measured via Western blots? If so, what was the outcome?

Response: Steady-state levels of autophagy proteins are difficult to interpret. Successful implementation of autophagy results in the degradation of P62/SQSTM and LC3. The level of mRNA for these genes is more proximal to the mechanistic event we hypothesize to mediate the effects of ApoE4. Regardless, we have performed a few Western-blot analyses, and the results are generally consistent with an inhibition of autophagy by ApoE4.

3. Possible to intracranially inject CBA2 into ApoE-TR4 vs ApoE-TR3 mice to validate its effect in the brain?

Response: This is a good suggestion. Since this is our proof-of-concept publication, we did not delve deeply into the mouse studies yet. CBA2 is predicted to have poor penetration of the blood-brain barrier; therefore, based on the pharmacophore modeling, we are generating new CBA2 analogs which have good BBB penetration and improved potency against ApoE4. More complete *in vivo* studies are underway.

4. Were T98G-E3 cells also treated with CBA2? Fig. 4e only had data of T98G-E4 treated with CBA2.

Response: Yes, we have treated T98G-E3 cells. As the report is already quite rich with data, we did not include these data. We have added this information as Supplemental Figure 3.

5. In Fig. 5b, why would E3+CBA2 also have a reduction, though not significant, still a 1/3 reduction? Seeing a reduction in the APOE3-expressing worms upon CBA2 treatment weakens the argument that CBA2 specifically blocks APOE4-induced A β aggregation in the *C.elegans* model.

Response: We certainly agree with the reviewer, based on our RT-PCR and Western blot analysis, we strongly believe that ApoE4 is the primary target for CBA2. In E3 cells, we did not notice any significant changes with autophagy gene expression when treated with CBA2. The notable reduction in aggregates (though not significant), seen in ApoE3 worms could be due marginal off-target effects of CBA2, especially in *C. elegans* environment. In order to make a transparent case, we chose to show results from both E3 and E4 worms.

6. On page 11, the statement “while *Sqstm1* was not among the individual genes significantly elevated in vivo, one of this gene’s transcriptional starts lacks any obvious CLEAR site; an alternative start manifests two candidates within 1 kb, and it is tempting to speculate that this may be the promoter that responded in cultured astrocytes” is difficult to follow. Which piece of data was this statement referring to?

Response: This speculative comment referred to the discrepancy between the response of *Sqstm1* in cultured astrocytes (Fig. 4D) versus *in vivo* (Fig. 5E). Of course, there are many reasons why a significant effect might not be detected in any given experimental paradigm. It is possible that the conditions which would induce *Sqstm1* in mouse liver were not met by a 20-hour fasting. But another possibility is that the gene’s two promoters are operational under different conditions and tissues. If the promoter wherein CLEAR sites are operational was induced by glucose-deprivation in astrocytes but not by fasting in liver, this could explain the discrepancy. Because the statement was entirely speculative, we have removed it.

7. Please consolidate all the statistical analyses into a single statistical analysis section located in the methods section. As currently written, information on statistical analyses appears variably in the figure legends, associated with specific analytical procedures, and briefly in the results section. Plus, the statistical analysis for your Fig.5e is hard to understand. Why was 3-way ANOVA performed?

Response: We have now consolidated all the statistical analysis into a single subsection under Methods. For the flow and easy understanding, we modified the figure legends accordingly.

General suggestions:

1. Throughout the manuscript, different terms “APOE ϵ 3,3”, “APOE ϵ 3, ϵ 3”, “APOE ϵ 4,4”, “APOE ϵ 4, ϵ 4” and so on were used when referring to APOE genotypes. The nomenclatures are inconsistent.

Response: We apologize for the inconsistencies in the terms, we now fixed the issues.

2. Please re-check the entire manuscript for typographical and formatting errors. E.g., on page 3 in the intro, “... AD-associated hallmark pathognomic aggregates of AD...”. On page 8, “sqst-1, lgg-2, and lmp-2 should be in uppercase”

Response: We apologize for the typos and formatting errors. We fixed the formatting errors in the manuscript. According to *WormBook: The Online Review of C. elegans Biology*, lowercase letters should be used to identify *C. elegans* genes, so those were left as per the original. We will defer to the editors' prerogative on this matter.

3. There are areas where words need to be added or corrected. For example, on page 7, there is a missing labeling of Fig in "... we found that only CBA2 treatment, not CBA1 or CBA3 (Fig a & c)...". On page 8, there is a wrong labeling of Fig in "... with CBA2 would improve their lysosomal autophagy and thus protect against A β aggregation (Fig 4g)".

Response: We corrected the noted errors throughout the manuscript.

4. Define ApoE-TR3 mice when it first appeared in the main text.

Response: We modified the statement where ApoE-TR3 first appears in the manuscript

5. Figure legends:

a. Specify your sample size throughout the different experiment

Response: We added our sample size in the figure legends as requested.

Fig. 1c, indicates what yellow regions are. Are they the residues that directly interact with CLEAR-DNA?

Response: Yes, we added a statement in the legend for Figure 1.

c. Fig. 3c, specifies the color-coded interactions. What is the purple, blue, and green region?

Response: We have now added the color keys in the figure to specify the type of interactions

Reviewer #2 (Remarks to the Author):

APOE4 is the strongest risk of late-onset AD and is associated with the autophagic/lysosomal failure in AD. In this manuscript, Balasubramaniam et al. employed a computational approach to identify small molecules that impede ApoE4::DNA binding. The leading compound CBA2 restored the expression of several autophagy genes in both in vitro and in vivo models and protected against A β 42 aggregation in *C. elegans* AD model. Overall, this study is novel and interesting, providing a new strategy to regulate autophagic/lysosomal function in AD.

Comments:

1. The structures of CBA1, 2, 3, 12, 23 and 30 should be presented in Fig. 2d. Fig. 2e shows 5 compounds, but the exact structures are not clear.

Response: We have a new section in Fig. 2 (Fig. 2f) which represent the 2D structure of CBA1, 2, 3, 12, 23 and 30.

2. The names of autophagy proteins are not consistent between Supplementary Figure 2b and Supplementary Figure 2c.

Response: We apologize for the mistake. We have now corrected Supplementary Figure 2b.

3. Numbers represent the migration of molecular weight markers (in kilodaltons) are missed in all western blot images.

Response: We apologize for not adding the molecular weight markers on the Western-blot images. We have now added MW for all.

4. CBA2 restored several autophagy genes in both in vitro and in vivo models. To strengthen the overall conclusion of this paper, the authors could perform autophagy assays to direct monitor autophagy in living cells after CBA2 treatment.

Response: We thank the reviewer for this is wonderful suggestion. As per the suggestion, we measured the autophagy in live (T98G) cells treated with or without CBA2; the results are presented as Figure 2, f & g, and in Supplementary Figure 3b &c. Results clearly substantiate our other data indicating that autophagy is thwarted in the presence of ApoE4. Furthermore, treatment CBA2 restores autophagy in T98G-E4 cells significantly, but does not affect T98G-E3 cells (Supplementary Figure 3b &c).

REVIEWERS' COMMENTS:

Reviewer #2 (Remarks to the Author):

The authors have addressed the comments that I raised before. I would recommend the manuscript for publication in Communications Biology.

One minor issue: In the right histogram of Fig 4h, should T98G-E3 be corrected to T98G-E4?

The authors have also addressed most comments that Reviewer 1 raised before.

Three minor issues:

1. Specific recommendations #5 of Reviewer 1: In Fig. 5b, why would E3+CBA2 also have a reduction, though not significant, still a 1/3 reduction? Seeing a reduction in the APOE3-expressing worms upon CBA2 treatment weakens the argument that CBA2 specifically blocks APOE4-induced A β aggregation in the C.elegans model.

The authors responded that "The notable reduction in aggregates (though not significant), seen in ApoE3 worms could be due marginal off-target effects of CBA2, especially in C. elegans environment". The authors should include such description in the discussion section to indicate that further studies are needed on the specificity of CBA2 on apoE4 in the future.

2. General suggestions # 1 of Reviewer 1: Throughout the manuscript, different terms "APOE ϵ 3,3", "APOE ϵ 3, ϵ 3", "APOE ϵ 4,4", "APOE ϵ 4, ϵ 4" and so on were used when referring to APOE genotypes. The nomenclatures are inconsistent.

It seems that the authors used APOE ϵ 4,4 and APOE ϵ 3,3 in the Discussion section now, but there is still inconsistency on terms of the ϵ 3 and ϵ 4 allele of APOE in the Introduction section (APOE ϵ 3,3, APOE ϵ 3, ϵ 3, APOE ϵ 4,4, APOE ϵ 4, ϵ 4). Please check again to make all necessary changes.

3. Please define "AD 3,3 and AD 4,4 patients" in the manuscript.

December 18, 2023

We thank the reviewers for handling our manuscript titled “**Rescue of ApoE4-related lysosomal autophagic failure in Alzheimer’s disease by targeted small molecules**”. As per the editorial checklist and reviewers’ comments, we have modified the manuscript accordingly for your consideration.

Response to the reviewers’ minor comments:

Reviewer #2 (Remarks to the Author):

The authors have addressed the comments that I raised before. I would recommend the manuscript for publication in Communications Biology.

One minor issue: In the right histogram of Fig 4h, should T98G-E3 be corrected to T98G-E4?

Response: We apologize for the typo, it is now corrected to T98GE4.

The authors have also addressed most comments that Reviewer 1 raised before.

Three minor issues:

1. Specific recommendations #5 of Reviewer 1: In Fig. 5b, why would E3+CBA2 also have a reduction, though not significant, still a 1/3 reduction? Seeing a reduction in the APOE3-expressing worms upon CBA2 treatment weakens the argument that CBA2 specifically blocks APOE4-induced A β aggregation in the C.elegans model.

The authors responded that “The notable reduction in aggregates (though not significant), seen in ApoE3 worms could be due marginal off-target effects of CBA2, especially in C. elegans environment”. The authors should include such description in the discussion section to indicate that further studies are needed on the specificity of CBA2 on apoE4 in the future.

Response: We have added the statement in the discussion

2. General suggestions # 1 of Reviewer 1: Throughout the manuscript, different terms “APOE ϵ 3,3”, “APOE ϵ 3, ϵ 3”, “APOE ϵ 4,4”, “APOE ϵ 4, ϵ 4” and so on were used when referring to APOE genotypes. The nomenclatures are inconsistent.

It seems that the authors used APOE ϵ 4,4 and APOE ϵ 3,3 in the Discussion section now, but there is still inconsistency on terms of the ϵ 3 and ϵ 4 allele of APOE in the Introduction section (APOE ϵ 3,3, APOE ϵ 3, ϵ 3, APOE ϵ 4,4, APOE ϵ 4, ϵ 4). Please check again to make all necessary changes.

Response: We checked the manuscript and fixed the inconsistencies

3. Please define “AD 3,3 and AD 4,4 patients” in the manuscript.

Response: We have defined AD 3,3 and AD 4,4 at the first occurrence.

As per the guidance, we have uploaded all the numerical source data, and original uncropped blot images to Mendeley data, and cited the DOI in the reference list.